# Weaning Ages Do Not Affect the Overall Growth or Carcass Traits of Hu Sheep

**DOI:** 10.3390/ani9060356

**Published:** 2019-06-14

**Authors:** Huiling Mao, Chong Wang, Zhongtang Yu

**Affiliations:** 1College of Animal Science and Technology, Zhejiang A & F University, Lin’an 311300, China; wangcong992@163.com; 2Key Laboratory of Applied Technology on Green-Eco-Healthy Animal Husbandry of Zhejiang Province, Hangzhou 311300, China; 3Department of Animal Sciences, The Ohio State University, Columbus, OH 43210, USA; yu.226@osu.edu

**Keywords:** carcass traits, growth, Hu sheep, ruminal development, weaning age

## Abstract

**Simple Summary:**

Early weaning increases the production efficiency of food animals. However, the success of early weaning is limited, in part, by the rate of ruminal volume and functional development, and the establishment of a functional rumen microbiome. Decades of research have focused on the effects of weaning age on the growth of sheep, but there is limited information on how weaning age affects ruminal development, fermentation profiles, and host biochemical and immune responses. Effect of weaning ages (30 vs. 45 d) of Hu lambs on the growth performance before weaning, rumen development when weaning, as well as carcass characteristics and meat quality after weaning were determined in this study. Our goal was to find if Hu sheep can be weaned younger than 45 d. The results of this study showed that there were no significant differences between the two weaning age groups on the growth performance, rumen development, slaughter characteristics, and meat quality, and thus weaning at d 30 could be recommended to save the feeding cost.

**Abstract:**

This study aimed to determine effects of weaning ages on growth, rumen development, and carcass characteristics and meat quality of Hu lambs. Thirty male Hu lambs were randomly divided into two weaning age groups: Weaned at 30 (W30) or 45 (W45) d of age. Blood samples were collected on the day of weaning before lambs (*n* = 5) were slaughtered, and then rumen sample was collected immediately after they were slaughtered. The intake of all feeds increased with age (*p* < 0.05), but were not affected by weaning age (*p* > 0.05). Oxidative stress indicators and immune variables, the plasma biochemical parameters did not differ between the two different weaning ages (*p* > 0.05). The two weaning age groups also had similar (*p* > 0.05) concentration of ruminal total volatile fatty acid. The two weaning age groups did not differ in body weight, carcass characteristics, or meat quality (*p* > 0.05) at d 120. These results indicate that weaning half a month earlier than the typical weaning age does not significantly affect the growth, ruminal development, or carcass characteristics of Hu lambs, and they can be weaned at 30 d of age to improve production efficiency.

## 1. Introduction

Sheep meat is an important meat product in many countries including Australia, India, New Zealand, Mongolia, and China. In today’s value-based marketing system, the sheep meat industry can benefit from raising lambs that produce high-quality meat products with reduced cost. Numerous studies have aimed to improve the growth and productivity in sheep [1,2,3]. Early weaning is one strategy to shorten the production time and to improve growth, feed efficiency [4], and carcass quality [5]. However, the success of early weaning is limited in part by the rate of ruminal volume and functional development [6] and establishment of a functional rumen microbiome [7]. Decades of research have focused on the effects of weaning age on the growth of sheep [3,6], but there is limited information on how weaning age affects ruminal development, fermentation profiles, and host biochemical and immune responses. 

The Hu sheep is the most important sheep breed raised in China because of its excellent prolificacy (3 to 4 lambs per parturition), rapid growth, and ability to adapt to poor quality feeds. However, the third and the fourth newborn lambs are always weaker than littermates and are underfed due to the limited milk from the dam. Consequently, the third and the fourth newborn lambs of a litter often suffer from undernutrition, retarded growth, increased risk of diseases, and mortality. To reduce mortality and increase production efficiency, Hu sheep producers typically feed milk replacers (MR) to newborn lambs, especially in the intensive sheep farms. Early weaning can reduce MR cost. Recently, much attention has been directed to early weaning of Hu sheep to help producers to lower production cost and increase profit [8]. In conventional feeding programs, Hu sheep are weaned at 45 d of age. The objective of this study was to determine if Hu sheep could be weaned younger than 45 d by examining how weaning at 30 d affects the growth, ruminal development, ruminal fermentation, blood chemistry, immune responses, carcass characters, and meat quality of Hu sheep.

## 2. Materials and Methods

### 2.1. Animals, Feeds and Experimental Design

All experimental protocols involving animals were approved by the Animal Use and Care Committee, Zhejiang A & F University, and the experimental procedures used in this study were in accordance with the university’s guidelines for animal research. The experiment was performed using Hu lambs at a breeding farm in the Zhejiang Province of China. 

Thirty healthy male Hu lambs at 5 d of age and with a mean body weight (BW) of 3.60 (±0.37 kg) were randomly divided into two weaning age groups: Weaning at d 30 (W30) or weaning at d 45 (W45), with 15 lambs in each group. Three lambs with similar BW within the same weaning group were placed in one indoor pen (1.0 × 1.0 m). All pens were located in the same stable, and both the treatments were grouped at one side of the stable. Individual pens were considered as experimental replicates (*n* = 5). All the lambs were subjected to a 5 d (from the age of 5 to 9 d) adaptation to the MR (150 g/liter water, Swot ASTCo., Ltd., Hangzhou, China; 945 g/kg dry matter (DM), and 184 g/kg crude protein (CP)), which was fed via baby bottles. The feeding trial began at the age of 10 d and ended at the 120 d.

Before weaning, all the lambs were also offered, *ad libitum*, starter pellets (914 g/kg DM, 174 g/kg CP, 169 g/kg neutral detergent fiber (NDF), and 73 g/kg acid detergent fiber (ADF)) and hay of Chinese wild rye (*Aneurolepidium Chinese Kitagawa*; 929 g/kg DM, 84.9 g/kg CP, 659 g/kg NDF, and 384 g/kg ADF) chopped into 6 to 8 cm lengths from d 10. To promote consumption of the starter pellets and hay of Chinese wild rye, MR was offered in decreasing amounts (30 mL decrease/d), from 780 mL/d (based on consumption in the adaptation period) at d 11 to 210 mL/d at d 30. Lambs in the W45 group continued to receive 210 mL/d MR from d 31 until weaning at d 45. The daily MR allowance was given in 3 equal portions at 0800, 1300, and 1800 h. 

On the weaning day (d 30 for the W30 group and d 45 for the W45 group), one lamb was randomly selected from each pen and slaughtered at a local slaughterhouse before the morning feeding to collect samples of ruminal content and tissue. Briefly, the lambs were stunned by captive bolt and exsanguinated according to the animal protection policy of China. Ruminal samples were collected from each slaughtered lamb and preserved (see below).

After weaning, the weaned lambs were fed a concentrate feed (882 g/kg DM, 151 g/kg CP, 396 g/kg NDF, and 219 g/kg ADF) and a forages mixture of bamboo shoot shell silage (966 g/kg DM, 102 g/kg CP, 701 g/kg NDF, and 364 g/kg ADF) and soybean straw (889 g/kg DM, 82.1 g/kg CP, 610 g/kg NDF, and 478 g/kg ADF), the ratio of bamboo shoot shell silage to soybean straw is 1 to 1. It took 5 days to make the change of diets. The concentrate feed and forages mixtures were offered in increasing amounts, 20 percentage replacement of pre-weaning diets per day. After that, lambs were fed post-weaning diets ad libitum. The concentrate feed and the forages mixture were offered separately at the same time. Feed orts were removed from bins and troughs at 0800 h each day, with new feed delivered three times daily at 0830, 1200, and 0430 h. All lambs had free access to drinking water. 

At d 120, one randomly selected lamb from each pen was slaughtered at a local slaughterhouse, as described above, to determine carcass characteristics and meat quality (see below).

The feeding trial was carried out at the same time as another study [9], and the animals in the group of W30 were the same as the control group in Mao et al. [9]. Therefore, the growth and productive data for control group W30 group were similar to those in the previous study [9].

### 2.2. Sample Collection and Performance Measurement

The amount of feed offered and refused and the BW were recorded for 2 consecutive days every ten days before weaning and then every fifteen days after weaning. Body weight was recorded before the morning feeding. 

On the weaning day (d 30 for W30 group and d 45 for W45 group), blood samples (approximately 10 mL) were collected from the jugular vein into heparinized tubes before lambs were slaughtered. Samples were then centrifuged as 3000× *g* at 4 °C for 15 min to obtain the plasma. Plasma samples were stored at −20 °C until subsequent analysis.

Immediately after slaughter (d 30 for W30 group and d 45 for W45 group), the rumen was cut open along the dorsal line and approximately 10 mL of ruminal fluid was collected and immediately stored at −20 °C until analysis for rumen fermentation characteristics. Rumen tissue (ventral fegion) samples, approximately 1.5 × 1.5 cm^2^ each, were collected from each animal and fixed in 10% formaldehyde until morphological measurement.

### 2.3. Blood Analysis

Concentration of total protein (TP), albumin, total glucose (TG), total cholesterol, low density lipoprotein cholesterol (LDLC), blood urea nitrogen (BUN), blood ammonia, total antioxidant capacity, malondialdehyde, H_2_O_2_, activities of superoxide dismutase, glutathione peroxidase (GSH-PX), and catalase were determined using commercial respective kits (Jiancheng, Nanjing, China). Concentrations of D-lactate, interleukin-6, cortisol, growth hormone, insulin-like growth factor, and tumor necrosis factor-α, were measured using respective enzyme-linked immunosorbent assay kits (Bangyi, Shanghai, China). 

### 2.4. Ruminal Fermentation Characteristics

About 2 g of each rumen content sample was added to 10 mL of sterile PBS (pH 7.0) and vortexed. After centrifugation at 12,000× *g* at 4 °C for 10 min, the supernatant was collected. To 1 mL of each supernatant, 20 μL of 85% orthophosphate acid was added and mixed by manually vortexing. The mixture was centrifuged again, as described above, and the final supernatant was subjected to gas chromatography to determine the concentrations of volatile fatty acids (VFAs) [10]. Ammonia-N concentration was determined, as described by Hu et al. [10]. Concentrations of the microbial protein were estimated using the purine method [11]. 

### 2.5. Ruminal Morphological Analysis

The rumen tissue samples were dehydrated through sequential washes in 50%, 70%, 80%, 95%, and 100% ethanol (30 min at each ethanol concentration), and then in 100% ethanol for 15 min. The dehydrated tissues were transferred to xylene for 2 h before being embedded in paraffin. Sections (5 μm in thickness) were stained with hematoxylin and eosin for observation under an optical microscope (Leica, Boston, MA, USA). The width and length of the ruminal papillae were measured and analyzed using the Motic Image Plus 2.0 software (Motic China Group Co. Ltd., Xiamen, China).

### 2.6. Analysis of Carcass Traits and Meat Quality

At the age of 120 days, the carcass traits and drip loss were evaluated the same as a previous study [9] after lambs were slaughtered (5 lambs per treatment). The girth rib (GR) value (the depth of muscle and fat tissue from the surface of the carcass to the lateral surface of the twelfth rib 110 mm from the midline) was directly measured using a GR knife [9].

### 2.7. Statistical Analyses

The results were statistically analyzed as a completely randomized design. Each pen with 3 lambs was considered one experimental unit for the analysis of feed intake (starter pellets and Chinese wild rye hay) and BW before weaning, while each randomly selected lamb was regarded as one experimental unit for other analyses. Growth data were analyzed using PROC MIXED procedure of SAS (SAS Inst. Inc., Cary, NC, USA). The model included weaning age, sampling age, and the weaning age sampling age interaction as fixed effects, lamb within weaning age as a random effect. The effect of sampling age was included as a repeated measure. Then, the data with no significance in the interaction were removed to allow an analysis of only the effect of weaning age and sampling age. Data on plasma variables, ruminal measurements, carcass traits, and drip loss, which were collected only once, were analyzed using the PROC GLM procedure of SAS. Statistical significance was declared at *p* ≤ 0.05 and trend at 0.05 ≤ *p* ≤ 0.10.

## 3. Results

### 3.1. Feed Intake and BW Gain

Feed intake of both the starter pellets and Chinese wild rye hay increased with age (*p* < 0.05) before weaning, and the two weaning age groups consumed a similar amount of starter pellets by d 30 (Figure 1). At d 30, the Chinese wild rye hay intake tended to be greater (*p* = 0.09) in the W45 than in the W30 groups. Weaning ages had no effect (*p* > 0.05) on total DM intake, CP intake or NDF intake (Figure 1c–e). After weaning, feed intake of both the concentrate feed and forage mixture increased with age (Figure 2, *p* < 0.05). The W30 group had a less (*p* < 0.05) intake of the concentrate feed at d 120 and a less intake of the forage mixture at d 105 (*p* < 0.05). At the other ages analyzed, the intake of both types of feeds was only numerically less in the W30 group than in the W45 group. The same variations were shown in total DM intake, CP intake, and NDF intake (Figure 2c–e).

The BW increased with age (Figure 3, *p* < 0.05) and no significant difference (*p* > 0.05) in BW was noted between the two weaning age groups until d 30 when the lambs in the W30 were weaned. The W30 group had numerically lower BW than the W45 group except at d 90 (*p* = 0.057) when the W30 group tended to have lower BW.

### 3.2. Plasma Variables

At weaning (at d 30 for W30 and at d 45 for W45), compared to the W45 group, the W30 group had a greater (*p* < 0.05) BUN concentration (Table 1). The W30 group tended to have a greater concentration of LDLC (*p* = 0.06) and D-lactate (*p* = 0.06), but the two weaning age groups were similar in other plasma biochemical parameters analyzed. 

### 3.3. Rumen Development and Fermentation Profiles

At weaning (at d 30 for W30 and at d 45 for W45), the lambs in the W45 group had a significantly longer (*p* < 0.05) rumen papillae and numerically wider papillae than those of the W30 group (Table 2). No significant differences (*p* > 0.05) in the concentration of total VFAs was noted between the two weaning age groups (Table 2). However, as compared to the W45 group, the W30 group had a lower molar proportion of acetate (*p* < 0.05), but a greater molar proportion of propionate (*p* < 0.05). The microbial protein in the ruminal content was greater (*p* < 0.05) in the W30 group than in the W45 group.

### 3.4. Carcass Traits and Meat Quality

Table 3 shows the carcass traits and meat quality of the two weaning age groups harvested at d 120. No significant difference (*p* > 0.05) was found in the carcass weight or dressing percentage between the W45 and the W30 groups. The two weaning ages also did not result in any significant difference (*p* > 0.05) in the meat quality measurements, including GR value or drip loss at either 24 h or 48 h.

## 4. Discussion

### 4.1. Feed Intake and BW Gain

Feed intake and BW gain are two important economic indicators in young and growing farm animals, including lambs, and thus they are usually evaluated in studies of growing animals. As shown in the present study, the two weaning ages, at 30 vs. 45 d, had little effect on feed intake, or BW after the weaning. This is in general agreement with the study using Ossimi lambs, which showed no difference in final live BW when weaned at d 56 vs. d 84 [6]. A similar finding was also reported by Aksakal et al. [12], who showed similar final live BW of Awassi lambs weaned at three different ages (45, 60, and 75 days). Furthermore, W30 lambs showed the same final live BW in comparison with W45 group lambs before weaning. This may be related to the same weaning practice, as all the lambs left their dam at d 5, fed the same amount of MR, and accessed feed freely. On the other hand, there were also no effect of weaning age on the concentration of TP and TG in plasma, which represent the metabolism of protein and glucose of animal, while glucose play an important role on the feed intake of preweaning calves [13]. These results suggest that Hu lambs can be weaned at d 30 without a significant adverse effect on animal growth. 

### 4.2. Rumen Development and Fermentation

Producers want to wean lambs as early as possible to improve production efficiency, and especially reduce the feed cost. However, lambs are born with a nonfunctional rumen and must rely on milk, or MR, to meet their nutrient requirements for growth and maintenance. A smooth transition from liquid to solid feed helps lambs adapt to solid feed and to cope with weaning stress [7], and this transition allows morphological and metabolic development of the rumen. Baldwin VI et al. [14] reported that proper starter pellets intake improved ruminal development. The early consumption of hay and concentrate feed stimulates the development of microbial fermentation and the functional development of the proventriculus [15,16,17]. In the present study, the lambs in both the weaning age groups were fed the starter pellets and hay of Chinese wild rye starting at 10 d of age, and the ruminal papillae grew and developed similarly in the two weaning age groups. In one study, early weaning was shown to increase the length and width of rumen papillae [16], and in another study, greater lengths and denser papillae in calves weaned at younger ages as compared to older ages [18]. However, Kehoe et al. [19] found no difference in the length or width of rumen papillae or the thickness of rumen wall when 124 Holstein heifer and bull calves were weaned at 21, 28, 35, or 42 d of age. In the present study, the W30 lambs had shorter ruminal papillae and slightly narrower papillae (only numerically). The discrepancy with the aforementioned studies may be attributed to the difference in ages (30 vs. 45 d) when the lambs were slaughtered and compared (in the aforementioned studies, all the calves were slaughtered at the same age). In future studies, lambs from different weaning age groups should be slaughtered at the same age and analyzed for comparison without the compounding effect of age difference. It should be pointed out that difference in animal species, and among them in animal breeds, cannot be excluded as another reason for the discrepancy.

The metabolism of ruminal epithelium changes as newborn ruminant animals grow, shifting from oxidation of glucose as the major energy source before weaning to oxidation of VFA as the main energy source after weaning [17,20,21]. Volatile fatty acids, especially butyrate, stimulate the development of ruminal papillaeand epithelia [15,22,23]. The two weaning age groups did not differ in total VFA concentration or concentration of individual VFAs, though the molar proportion of acetate and propionate differed, with lower acetate molar proportion and greater propionate molar proportion in the W30 lambs. This is consistent with the rather similar feed intake. These results indicate that weaning Hu lambs 15 d earlier than the typical weaning age (45 d) does not adversely affect the ruminal development. 

The W30 group had a numerically greater ruminal ammonia concentration than the W45 group, but the former had a greater content of microbial protein than the latter. This might indicate an improved dietary N utilization in the early weaned animals. In a previous study, we found that early-weaned calves had a lower ruminal ammonia concentration, while numerically greater content of microbial protein than those weaned at typical ages (120 d) [24]. Abou Ward found that early-weaned lambs (d 56) had greater N utilization due to the well-developed rumination and fermentation in comparison with the lambs weaned at a latter age (d 84) [6]. In the early studies of ruminal nitrogen metabolisms, Hungate [25] believed that microbial hydrolysis of dietary N increased the ruminal ammonia concentration, while the increased ammonia utilization of rumen microbiome [26] and the increased ability of the rumen to absorb ammonia [27] could decrease the ruminal ammonia concentration. Nevertheless, weaning Hu lambs 15 days earlier than the typical weaning program may not negatively affect the nitrogen metabolism in the rumen or the microbial protein supply to the small intestine. This premise is supported by the similar BW between the two weaning age groups. The effect of weaning age on the ruminal microbiome and nitrogen metabolism can be investigated in future studies using contemporary omics technologies, such as metagenomics and metabolomics. 

### 4.3. Carcass Traits and Meat Quality 

Carcass weight and traits are the ultimate indicators of meat-producing animal values. In a recent study, it was shown that carcass traits of Hu lambs were mainly affected by dietary nutrition level [28], rather than weaning ages. As indicated by the similar carcass weight and traits between the two weaning age groups, consistent with the similar live BW, Hu lambs can be weaned at age of 30 without a negative impact on carcass yield or traits. 

## 5. Conclusions

Based on the comparison of multiple measurements, including feed intake, animal growth, ruminal development, ruminal fermentation, blood measurements, and carcass characteristics, weaning of Hu lambs at age of 30 can be used to increase the production efficiency and save cost on feed.

## Figures and Tables

**Figure 1 animals-09-00356-f001:**
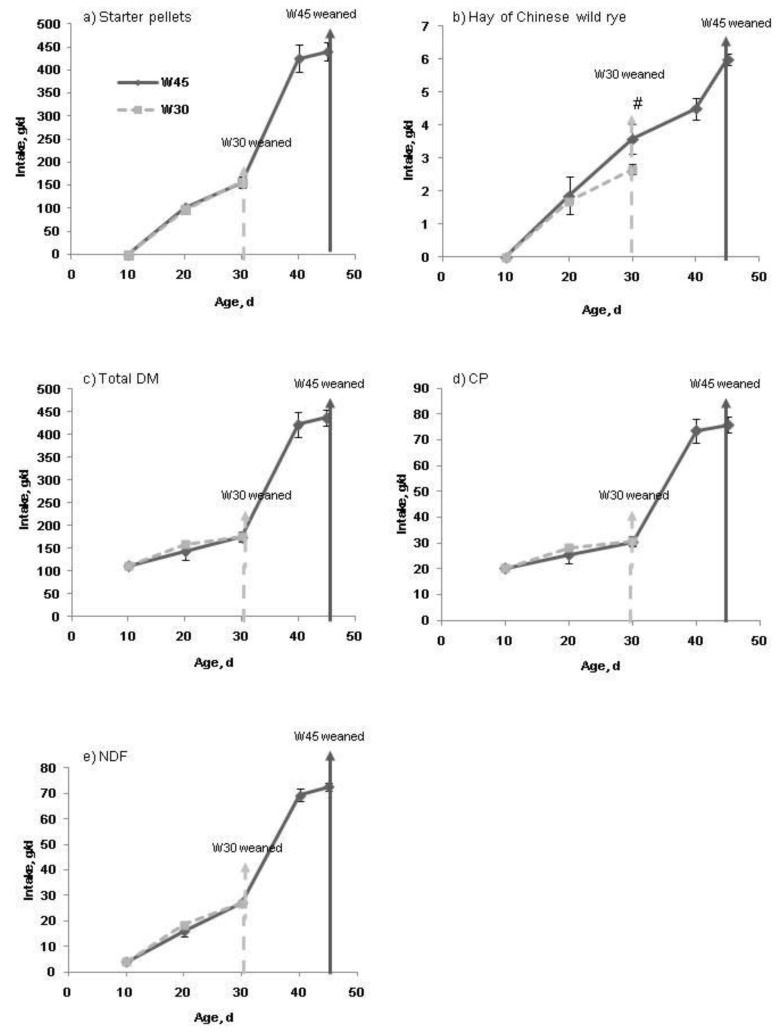
Intake of starter pellets (**a**); hay of Chinese wild rye hay (**b**); total dry matter (DM, (**c**)); crude protein (CP, (**d**)); and neutral detergent fiber (NDF, (**e**)) at different ages of the growing lambs before weaning. Dashed line, lambs weaned at d 30 (W30); Solid line, lambs weaned at d 45 (W45). Error bars indicate SE. # 0.05< *p* < 0.10.

**Figure 2 animals-09-00356-f002:**
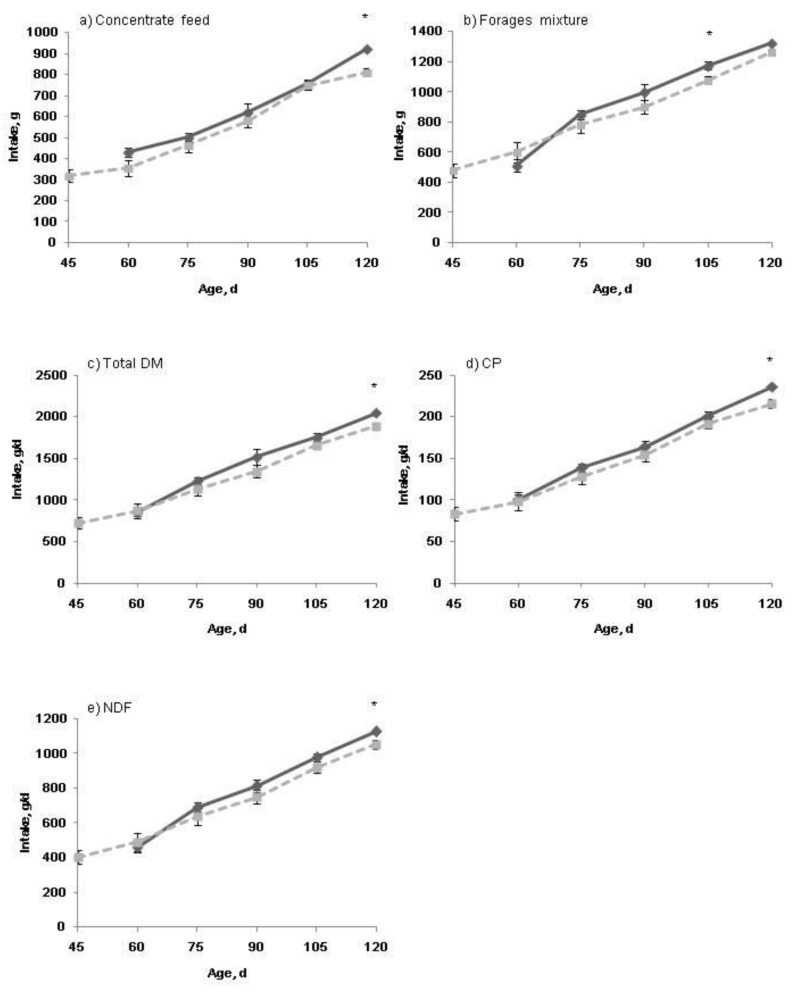
Intake of concentrate feed (**a**); forage mixture (**b**); total dry matter (DM, (**c**)); crude protein (CP, (**d**)); and neutral detergent fiber (NDF, (**e**)); at different ages of the growing lambs after weaning. Dashed line, lambs weaned at d 30 (W30); Solid line, lambs weaned at d 45 (W45). Error bars indicate SE. * *p* < 0.05.

**Figure 3 animals-09-00356-f003:**
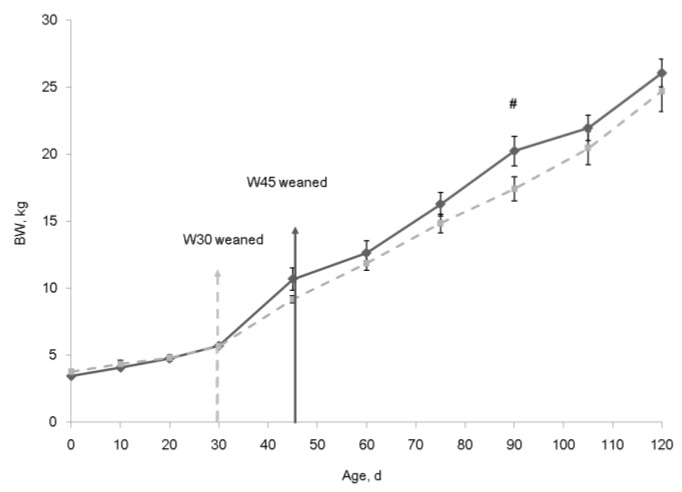
Cumulative BW at different ages of the growing lambs. Dashed line, lambs weaned at d 30 (W30); Solid line, lambs weaned at d 45 (W45). Error bars indicate SE. # 0.05 < *p* < 0.10.

**Table 1 animals-09-00356-t001:** Concentrations of plasma measurements in growing lambs at weaning (at d 30 for W30; at d 45 for W45) ^a^.

Variables ^b^	Treatments ^c^		SEM	*p*-Value
	W30	W45		
**Biochemical**				
TP, g/L	38.0	48.2	3.75	0.113
ALB, g/L	23.2	27.6	3.98	0.480
TG, mmol/L	1.09	1.72	0.322	0.229
T-CHO, mmol/L	2.12	1.74	0.482	0.618
LDL-C, mmol/L	2.38	0.61	0.497	0.064
BUN, mg/L	367	181	36.7	0.027
Blood ammonia, µmol/L	196	159	21.8	0.769
**Antioxidant**				
T-AOC, U/mL	3.95	3.91	0.647	0.452
SOD, U/mL	33.4	42.3	3.22	0.107
GSH-PX, U/mL	201	317	22.9	0.270
MDA, nmol/mL	4.55	5.48	1.002	0.342
CAT, U/mL	1.08	3.88	0.548	0.358
**Immune parameters**				
D-lactate, ng/mL	1024	950	20.5	0.064
Interleukin-6, pg/mL	143	152	4.76	0.256
Cortisol, ng/mL	33.2	33.0	1.97	0.924
Growth hormone, ng/mL	20.9	22.2	0.59	0.184
IGF, ng/mL	90.4	92.2	4.35	0.763
TNF-α, ng/mL	307	305	6.0	0.821

^a^ Number of male Hu lambs per treatment = 5; ^b^ TP = total protein; ALB = albumin; TG = total glucose; T-CHO = total cholesterol; LDLC = low density lipoprotein cholestero; BUN = blood urea nitrogen; T-AOC = total antioxidant capacity; SOD = superoxide dismutase; GSH-PX = glutathione peroxidase; MDA = malondialdehyde; CAT = catalase; IGF = insulin-like growth factors; TNF-α = tumor necrosis factor-α; ^c^ W30 = weaned at d 30; and W45 = weaned at d 45.

**Table 2 animals-09-00356-t002:** Measurement of rumen papillae and fermentation profiles of growing lambs at weaning (at d 30 for W30; at d 45 for W45) ^a^.

Variables	Treatments ^b^		SEM	*p*-Value
	W30	W45		
Papillae length, µm	581	985	67.2	0.017
Papillae width, µm	323	369	32.0	0.369
VFA Concentration, mg/g				
Total	1.58	2.50	0.370	0.203
Acetate	0.72	1.52	0.235	0.097
Propionate	0.82	0.94	0.182	0.715
Butyrate	0.04	0.05	0.005	0.581
VFA Molar proportion, mM/100 mM				
Acetate	53.0	69.2	2.07	0.011
Propionate	45.0	29.5	2.12	0.013
Butyrate	1.93	1.29	0.217	0.160
Ammonia N, mg/L	6.51	3.66	0.988	0.117
Microbial protein, mg/ml	17.4	14.1	0.70	0.045

^a^ Number of male Hu lambs per treatment = 5; ^b^ W30 = weaned at d 30; W45 = weaned at d 45.

**Table 3 animals-09-00356-t003:** Carcass characteristics and meat quality of lambs slaughtered at age of 120 d ^a^.

Variables	Treatments ^b^		SEM	*p*-value
	W30	W45		
**Carcass characteristics**				
Body weight, kg	23.7	26.4	1.35	0.190
Carcass weight, kg	10.0	11.3	0.59	0.160
Dressing percent, %	42.2	42.8	0.56	0.718
**Meat quality**				
GR, cm	1.08	1.02	0.113	0.477
**Drip loss**				
24 h	7.55	5.55	0.680	0.272
48 h	10.3	8.13	1.244	0.277

^a^ Number of male Hu lambs per treatment = 5; ^b^ W30 = weaned at d 30; W45 = weaned at d 45.

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
