# Peer review of "Weaning Ages Do Not Affect the Overall Growth or Carcass Traits of Hu Sheep"

_animals, 2019, doi:10.3390/ani9060356_

Round 1
Reviewer 1 Report
You can find an attached file

Author Response
General consideration
The paper looked through different weaning ages on growth performance, rumen development fermentation profiles and host biochemical and immune response. Nevertheless, some criticism came up mainly on experimental design. Indeed, authors, in the introduction and discussion sections explained as early weaning is of strong importance to guarantee an improvement of dam production efficiency and shorten their postpartum resumption of reproduction, but in the trail authors used lambs fed milk replaced before weaning explaining as this strategy is usually used to feed the third and fourth newborn lambs. In my opinion authors should better explain the aim of the trial otherwise they should have used dam sucking lambs weaned at different ages.
Author response: Improving the dam production efficiency and shorten their postpartum resumption of reproduction is one merit of early weaning, so we presented it in the introduction. As you said, we used milk replacer to feed lambs, the results of this study did not refract this merit, thus this sentence in introduction was removed and the discussion part was revised.
Finally, I suggest major revision as final decision.
Author response: We appreciate the critical but constructive comments and suggestions. We addressed all the comments when we revised the manuscript.
Structure
Abstract
Results are summarized in a sounding manner, although I suggest to better specify in the aim of the trial that studied lambs are weaned using milk replacer and came from triplet or more lambing as specified in the introduction.
Author response: As mentioned above, we revised the introduction part.
1. Introduction
Introduction describes in a satisfactory manner the status of the art on the topic of the paper but the aim of study is not so clear, in particular authors reported as in the Hu sheep herd is possible to find dam sucking lambs and milk replacer fed lambs without specify if the experiment involved dam sucking lambs or milk replacer fed lambs. I suggest to improve and clarify this part because in Material and methods authors specified as the experiment involved milk replacer fed lambs.
Author response: As mentioned above, we revised the introduction part.
2. Material and methods
Materials and methods is well organized, although the section 2.3 Animal and diet should be improved.
Author response: The section 2.3 was revised.
2.1 Animals, feeds and experimental design
This part should be improved as specified in the Particular comments section
Author response: Response presented in Specific comments.
2.2 Growth parameters Measurement and Sample Collection
Slaughter procedure and the number of animal and age at slaughtering for each treatment should be described, indeed, in the previous section "animals, feeds and experimental design" authors described in a general manner the experimental design.
Author response: Thanks for your suggestion. This part has been revised (see section 2.2)
2.3 Blood sample collection and analysis
See the section Specific comments
Author response: Response presented in Specific comments.
2.4 Ruminal Fermentation Characteristics
No comments
2.5 Ruminal Morphological Analysis
No comments
2.6 Analysis of carcass traits and meat quality
The English form should be improved
Author response: This part was revised (see the section 2.6).
2.7 Statistical analysis
No comments
3 Results
3.1 Feed intake and BW gain
Hay DMI showed in Fig.1 ranged from 2 to around 6g/head and day till weaning and starts from around 400g/head and day immediately after weaning (figure 2): please explain. More in general for a better understanding of the paper I suggest to insert a table in which are reported the analysis of variance results concerning the effects of weaning method and age or weighing day on daily DMI and BW and their interaction.
Author response: After weaning, the diets of animals were changed. We spend 5 days to make the change of diets. The concentrate feed and forages mixture were offered in increasing amounts, 20 percentage replacement of pre-weaning diets per day. After that, lambs were fed post-weaning diets ad libitum. The forages mixture intake, which showed in figure 2, was measured 15 days after weaning (d45 for W30, d 60 for W45). On the other hand, lambs needed to take more solid feed to meet their requirements because they have no milk replacer feeding.
Animals were weaned at different days and we measured the feed intake every ten days, so there were no starter and Chinese wild rye intake data for W30 on d 40 or d 45, thus we think the figures may be more visual.
3.2 Plasma Variables
No comments
3.3 Rumen development and Fermentation profiles
No comments
4. Discussion
4.1 Feed intake and BW Gain
This section should be improved taking in account of the results reported in figures 1 and 2.
Author response: The more discussion were added to this section (see section 4.1).
4.2 Rumen development and fermentation
No comments
4.Conclusions
Conclusions are consistent with results achieved.
Specific comments
1. Introduction
Line 49:"...major sheep species..." I suggest"...most important sheep breed..."
Author response: Revised as suggested.
2. Materials and Methods
2.1 Animal, feeds and experimental design
Line 102 to Line 105: I suggest to improve this part.
Author response: This part was revised.
2.2 Blood sample collection and analysis
Line 115 to Line 117: I suggest to modify the sentence in "... and plasma parameters were analysed according to Mao et al.2019"
Author response: Follow another reviewer's suggestion, this part was revised (see section 2.3).
3. Results
No specific comments
4. Discussion
4.2 Rumen Development and Fermentation
Line 210 to Line 211: I suggest to modify the sentence in "It should be pointed out that difference in animal species and among them in animal breeds cannot be excluded as another reason for the discrepancy"
Author response: Thanks for your suggestion, the sentence was revised.
Line 215:" rumen papillaeand epithelia..." I suggest to modify in " rumen papillae and epithelia"
Author response: Revised as suggested.
Line 222 to Line 223:"This apparent contradiction is enigmatic", I suggest to modify the sentence: It appear as emotion expression, authors should try to justify results achieved.
Author response: Thanks for your comments. We revised this part.
Line 223:"in one previous study..." I suggest "in a previous study.."
Author response: Revised as suggested.
Line 225 to Line 231: This speculation sounds confusing I suggest to better explain what authors should say.
Author response: Sorry to make the confusing. The reduced ammonia concentrations in the early weaned lambs might indicate an improved dietary N utilization for the well developed rumination and fermentation. Microbial hydrolysis of dietary N increased the ruminal ammonia concentration, while the increased ammonia utilization of rumen microbiome and the increased ability of the rumen to absorb ammonia could decrease the ruminal ammonia concentration. This part was revised.

Reviewer 2 Report
General comments:
This paper is in general well written and scientifically sound. Additional details about the methodology and additional results have to be added (as specified bellow) to improve the paper.
Specific comments:
Summary (line 10) and introduction (line 44): Both in the summary and in the introduction the authors mentions the rate of ruminal development. It is not clear what is meant by “the rate”, is it volume, functionality or both?
Materials and methods:
Line 73: please give more information about the pen organization: were all pens located in the same stable, was an alternation of treatment used between consecutive pens or were the same treatments grouped at one side of the stable?
Line 75: Please give the concentration of milk replacer (g/liter water or g/liter milk)
Line 78-79: What was the energy content of the starter pellets and the hay?
Line 85: Was feed, milk or water provided on the day of slaughter?
Line 98: are 3 feeding times a day in agreement with common practices?
Line 107: In line 97 is mentioned that orts were removed daily. Why was the daily feed intake not calculated per day?
Line 110/119: Rumen sampling: Were solids (fibers), liquids or both sampled? What was the volume of the rumen sample?
Line 112: What region of the rumen was samples (dorsal, ventral,… maybe a picture or drawing can illustrate the sample location? It is known that there are regional differences in rumen wall development, papillae density and papillae development.
Line 116: Please describe the plasma parameters that were analysed in this study. At this point, the reader will not look into the reference paper to known the analysed parameters. For methodology it is OK to refer to another paper.
Line 135: please list the analysed carcass and meat quality parameters
Results:
Line 116/163: please explain the abbreviations in the text, not only in the table.
Table 2: Was papillae density measured?
Figure 1: Additional information on total dry matter intake, total energy intake, feed and nitrogen efficiency between treatments has to be included in the results section and tables or figures.
Discussion:
Line 185: The weaning age is much later in the publications the authors refer to. Is this due to the breed differences?
Line 207-210: I strongly agree with the authors that it would have been (more) useful if lambs of both treatments would have been slaughtered at both 30 and 45 days, allowing to make conclusions about the age effect.
Line 212: Additional information about rumen functionality parameters would have been informative (gene expression or staining of rumen wall samples for specific proteins expressed in the rumen wall).
Author Response
Response to Reviewer 2 Comments
General comments:
This paper is in general well written and scientifically sound. Additional details about the methodology and additional results have to be added (as specified bellow) to improve the paper.
Author response: Thanks for your encouragement and criticizing comments.
Specific comments:
Summary (line 10) and introduction (line 44): Both in the summary and in the introduction the authors mentions the rate of ruminal development. It is not clear what is meant by “the rate”, is it volume, functionality or both?
Author response: Thanks for the comments. This "rate" means the ruminal development rate of both the volume and function. This sentence was revised.
Materials and methods:
Line 73: please give more information about the pen organization: were all pens located in the same stable, was an alternation of treatment used between consecutive pens or were the same treatments grouped at one side of the stable?
Author response: All indoor pens were located in the same stable, and both the treatments grouped at one side of the stable. This information was added to the revised manuscript.
Line 75: Please give the concentration of milk replacer (g/liter water or g/liter milk)
Author response: The concentration of MR was 150g/liter water.
Line 78-79: What was the energy content of the starter pellets and the hay?
Author response: The chemical composition of the starter pellets and the hay were analyzed in the laboratory, but we did not analyze the energy content of the feed.
Line 85: Was feed, milk or water provided on the day of slaughter?
Author response: No, the animals were slaughtered before the morning feeding. This information was added to the revised manuscript.
Line 98: are 3 feeding times a day in agreement with common practices?
Author response: Yes, 3 feeding times a day in agreement with common practices. We usually use 2 or 3 feeding times a day, because the animals used in this study are lambs, we chose 3 feeding times a day in this study.
Line 107: In line 97 is mentioned that orts were removed daily. Why was the daily feed intake not calculated per day?
Author response: We just removed the orts daily, but have not measure the weight of orts, and we think the workload was too big if we measured the weight of orts every day.
Line 110/119: Rumen sampling: Were solids (fibers), liquids or both sampled? What was the volume of the rumen sample?
Author response: About 10 ml of rumen fluid sample were collected, this information was added to the manuscript.
Line 112: What region of the rumen was samples (dorsal, ventral,… maybe a picture or drawing can illustrate the sample location? It is known that there are regional differences in rumen wall development, papillae density and papillae development.
Author response: Yes, you are right, there are regional differences in rumen wall development, rumen papillae density and development. However, as reported by Yang et al., (2015), who measured the rumen papillae length and width of left, right and ventral part of rumen, there were no significant difference among different regional since 4 week age of lambs. Thus, we just collected and measured the ventral part of rumen in this study
Line 116: Please describe the plasma parameters that were analysed in this study. At this point, the reader will not look into the reference paper to known the analysed parameters. For methodology it is OK to refer to another paper.
Author response: That's a good suggestion, and this part was revise (see section 2.3).
Line 135: please list the analysed carcass and meat quality parameters
Author response: This part was revised (see section 2.6).
Results:
Line 116/163: please explain the abbreviations in the text, not only in the table.
Author response: Thanks for your suggestion, this was revised.
Table 2: Was papillae density measured?
Author response: No, we did not measure the papillae density.
Figure 1: Additional information on total dry matter intake, total energy intake, feed and nitrogen efficiency between treatments has to be included in the results section and tables or figures.
Author response: We have added the DMI, CP intake and NDF intake data to Fig.1 and Fig.2.
Discussion:
Line 185: The weaning age is much later in the publications the authors refer to. Is this due to the breed differences?
Author response: Yes, we believed the breed differences was the major reason.
Line 207-210: I strongly agree with the authors that it would have been (more) useful if lambs of both treatments would have been slaughtered at both 30 and 45 days, allowing to make conclusions about the age effect.
Author response: Thank you for agreeing with us, we will do better in the future studies.
Line 212: Additional information about rumen functionality parameters would have been informative (gene expression or staining of rumen wall samples for specific proteins expressed in the rumen wall).
Author response: This is a good suggestion. The effect of weaning age on the rumen microbiome and nitrogen metabolism will be investigated in future studies using contemporary -omics technologies, such as metagenomics and metabolomics.

Round 2
Reviewer 1 Report
comments are in the enclosed file

Author Response
Response to Reviewer 1 Comments
We appreciate the comments and suggestions, the point-to-point responses are shown below.
Specific comments
1. Introduction
Line 48:"...sheep meet industry..." should be changed in"...sheep meat industry..."
Author response: The typo was corrected.
2. Materials and Methods
2.3 Blood analysis
Line 152: "... and catalase determined..." should be changed in "... and catalase were determined ..."
Author response: Revised as suggested.
2.4 Ruminal Fermentation Characteristics
Line 158:" 2.4 Ruminal...should be changed in "2.4 Ruminal...
Author response: Revised as suggested.
3. Results
4. Discussion
4.1 Feed intake and BW Gain
Line 255:"...metabolism of glucose and protein of animal..." should be change in "... Metabolism of protein and glucose of animal..."
Author response: Revised as suggested.
Line 256:"... Feed intake of preweaning claves" should be changed in "... feed intake of preweaning calves[13]"
Author response: The typo was corrected
4.2 Rumen Development and Fermentation
Line 299: "...later..." I suggest "... latter".
Author response: Revised as suggested.
